# Excellent Energy Storage Performance in Bi(Fe_0.93_Mn_0.05_Ti_0.02_)O_3_ Modified CaBi_4_Ti_4_O_15_ Thin Film by Adjusting Annealing Temperature

**DOI:** 10.3390/nano12050730

**Published:** 2022-02-22

**Authors:** Tong Liu, Wenwen Wang, Jin Qian, Qiqi Li, Mengjia Fan, Changhong Yang, Shifeng Huang, Lingchao Lu

**Affiliations:** 1Shandong Provincial Key Laboratory of Preparation and Measurement of Building Materials, University of Jinan, Jinan 250022, China; shandongmems@163.com (T.L.); wangwenwen_0717@163.com (W.W.); j28_qian@163.com (J.Q.); q1635376692@163.com (Q.L.); fan15176103198@163.com (M.F.); mse_huangsf@ujn.edu.cn (S.H.); mse_lulc@ujn.edu.cn (L.L.); 2MEMS Institute of Zibo National High-Tech Development Zone, Zibo 255000, China

**Keywords:** CBTi-BFO, fine grain, electric breakdown strength, recoverable energy storage

## Abstract

Dielectric capacitors with ultrahigh power density are highly desired in modern electrical and electronic systems. However, their comprehensive performances still need to be further improved for application, such as recoverable energy storage density, efficiency and temperature stability. In this work, new lead-free bismuth layer-structured ferroelectric thin films of CaBi_4_Ti_4_O_15_-Bi(Fe_0.93_Mn_0.05_Ti_0.02_)O_3_ (CBTi-BFO) were prepared via chemical solution deposition. The CBTi-BFO film has a small crystallization temperature window and exhibits a polycrystalline bismuth layered structure with no secondary phases at annealing temperatures of 500–550 °C. The effects of annealing temperature on the energy storage performances of a series of thin films were investigated. The lower the annealing temperature of CBTi-BFO, the smaller the carrier concentration and the fewer defects, resulting in a higher intrinsic breakdown field strength of the corresponding film. Especially, the CBTi-BFO film annealed at 500 °C shows a high recoverable energy density of 82.8 J·cm^−3^ and efficiency of 78.3%, which can be attributed to the very slim hysteresis loop and a relatively high electric breakdown strength. Meanwhile, the optimized CBTi-BFO film capacitor exhibits superior fatigue endurance after 10^7^ charge–discharge cycles, a preeminent thermal stability up to 200 °C, and an outstanding frequency stability in the range of 500 Hz–20 kHz. All these excellent performances indicate that the CBTi-BFO film can be used in high energy density storage applications.

## 1. Introduction

At present, energy and environmental issues are the focus of social attention. The vigorous development of green and clean energy (such as wind and solar energy) is one of the future trends. The instability and intermittency of green energy put forward higher requirements for energy storage technology [1,2,3,4,5]. Dielectric capacitors typically display ultrafast charge–discharge rates and long life-time, temperature/frequency stability, fatigue resistance, which play key roles in various modern electrical and electronic systems, such as hybrid electric vehicle, aircraft and military [6,7,8,9]. Film capacitors offer a smaller size and higher energy storage density, making them easier to integrate into circuits than other devices such as ceramic capacitors [10]. Currently, most commercial dielectrics are mainly made of organic polymers, such as biaxially oriented polypropylene (BOPP), which have been widely used as the dielectric layer in power inverter capacitor systems, making the storage system bulky due to the low energy density (<5 J·cm^3^). Furthermore, the operating temperature of BOPP cannot be higher than 80 °C, which increases the difficulty of structure design due to the need for an extra cooling system [11,12]. By contrast, inorganic dielectric film capacitors have the advantages of relatively high energy densities, better thermal stability in wider operating temperature ranges, and long-term endurance. Among this, inorganic ferroelectric film capacitors (TFFCs) are considered as good candidates for energy storage due to their large polarization and high temperature resistance [13,14]. However, low energy storage density and efficiency limit its further development in energy storage applications; thus, further improvements are needed.

For film capacitors, two important energy storage parameters, the recoverable energy storage density (*W*_rec_) and energy storage efficiency (*ƞ*), can be calculated from the measured hysteresis loops adopting the following equations [15,16]:(1)$$W_{\mathrm{rec}}=\int_{P_{r}}^{P_{m}}E\;dP\;$$
(2)$$W_{t}=\int_{0}^{P_{m}}E\;dP\;$$
(3)$$ƞ=\frac{W_{\mathrm{rec}}}{W_{t}}\times 100\%\;$$
where *E*, *W*_t_, *P*_m_ and *P*_r_ are the applied electric field, total energy storage density, the maximum polarization and remanent polarization during the discharge process, respectively. Therefore, *W*_rec_ can be improved by increasing the difference between *P*_m_ and *P*_r_, and the electric breakdown strength (*E*_b_). It is well known that the *E*_b_ of dielectric materials is mainly contingent on its microstructure, such as grain size and degree of densification. Therefore, increasing *E*_b_ by reducing grain size is an effective way to improve energy storage performance [17]. Wang et al. sort out the relationship between grain size and electric breakdown strength, confirming the optimization effect of energy storage via grain size-engineering [6]. As is well known, the annealing temperature has an immense impact on the quality of films prepared by chemical solution deposition (CSD). For instance, Wang et al. unveil a large value of *W*_rec_ up to 91.3 J·cm^−3^ at 4993 kV·cm^−1^ for Pb_0.88_Ca_0.12_ZrO_3_ (PCZ) antiferroelectric thin films by designing a nanocrystalline structure of the pyrochlore phase by optimizing the annealing temperature to 550 °C [18]. However, the negative effect caused by the application of lead-containing dielectrics to human health and environmental sustainability cannot be ignored, and the exploration of lead-free energy storage materials is raised in the agenda. For example, Zuo et al. investigate that a high *W*_rec_ of 8.12 J·cm^−3^ and a great *ƞ* of ∼90% are obtained simultaneously in BiFeO_3_-BaTiO_3_-NaNbO_3_ ceramics, which can be attributed to the significantly enhanced *E*_b_ of BiFeO_3_-based ternary solid solutions originating from the increased resistivity and refined grain size [19].

Bismuth layer-structured ferroelectric (BLSF) compounds, such as SrBi_2_Nb_2_O_9_ (SBN), SrBi_2_Ta_2_O_9_ (SBT), Bi_4_Ti_3_O_12_ (BIT), CaBi_4_Ti_4_O_15_ (CBTi), belong to a large category of ferroelectric materials [13,20,21,22]. They have the advantages of excellent anti-fatigue property, large dielectric constant and small dielectric loss, high resistivity and low leakage current density, high ferroelectric Curie transition temperature, and so on [23,24,25,26]. Those traits show a good application prospect in the field of dielectric energy storage, but there is little research on BLSF compounds in this field [27,28]. This is mainly due to their intrinsic shortcomings, namely, relatively low polarization and high coercive field, which lead to lower energy density and higher losses in energy storage applications [29]. Recently, Pan et al. presented a composition modification method in ferroelectric Aurivillius Bi_3.25_La_0.75_Ti_3_O_12_ by introducing BiFeO_3_ to increase the polarization value and optimize hysteresis loops, in which *W*_rec_ (113 J·cm^−3^) and *ƞ* (80.4%) are observed. Yang et al. prepared a series of 0.6BaTiO_3_-0.4Bi_3.25_La_0.75_Ti_3_O_12_ thin films, and the modified thin film also shows higher dielectric breakdown strength and polarization. CBTi is also a representative BLSF compound, which exhibits distinct advantages including being lead-free and fatigue-free. Meanwhile, it possesses a high Curie point of about 790 °C to be used in relatively high temperature applications [30]. However, it also faces troubles of low spontaneous polarization.

In this work, we select Bi(Fe_0.93_Mn_0.05_Ti_0.02_)O_3_ introduced into CBTi, namely, CBTi-BFO, to reduce leakage current and enhance breakdown field strength. In order to further optimize the energy storage performance of CBTi-BFO thin films, the effect of annealing temperature on their energy storage capacity has been studied in detail. We found that the microstructures of the CBTi-BFO thin films can be dominated by adjusting the annealing temperature. The CBTi-BFO film annealing at 500 °C possesses an excellent *W*_rec_ of 82.8 J·cm^−3^ and *ƞ* of 78.3%, simultaneously, due to the obviously enhanced *E*_b_ of 3596 kV·cm^−1^. Meanwhile, the film shows outstanding temperature/frequency stability up to 150 °C and superior fatigue stability after 10^7^ switch cycles. The findings overcome the shortcomings of organic thin films in energy storage, including low energy storage density and low application temperature, unveiling an effective way towards high performance lead-free and eco-friendly ferroelectric materials for energy storage applications.

## 2. Materials and Methods

Fabrication: CBTi-BFO films were synthesized on Pt/Ti/SiO_2_/Si substrates by chemical solution deposition. Bismuth nitrate pentahydrate [Bi(NO_3_)_3_·5H_2_O], calcium nitrate tetrahydrate [Ca(NO_3_)_2_·4H_2_O], iron nitrate nonahydrate [Fe(NO_3_)_2_·9H_2_O], manganese acetate tetrahydrate [C_4_H_6_MnO_4_·4H_2_O] as raw materials were dissolved in ethylene glycol and acetic acid. Here, 10 mol% excess Bi was added to compensate for elements volatilization. After that, the tetrabutyl titanate and acetylacetone were added into the mixed clarified salt solution. The final concentration of the precursor solution was 0.1 M. After 24 h of aging, the precursor solution was spin coated on Pt/TiO_2_/SiO_2_/Si substrates with a speed of 4000 rpm for 30 s. After that, the as-prepared CBTi-BFO films were pyrolyzed at 350 °C for 120 s and annealed at 450, 500, 550, 600 °C for 10 min in a rapid thermal annealing procedure, respectively. The spin coating and annealing process procedures were duplicated up till the desired thickness of 300~700 nm was obtained. Circular Pt top electrodes, ~200 μm in diameter, were sputtered through a shadow mask on the films for the next electrical measurements.

Characterization and Measurements: The crystalline structure of CBTi-BFO films was characterized by an X-ray diffractometer (XRD) with Cu K*α* radiation (XRD, D8 ADVANCE, Karlsruhe, Germany). The cross-sectional microstructure and surface morphology were characterized by a field emission scanning electron microscope (FESEM, ZEISS Gemini300, Oberkochen, Germany). The polarization-electric field (*P-E*) loops and insulating characteristic were acquired from a standard ferroelectric tester (aixACCT TF3000, Aachen, Germany). The frequency-dependent dielectric properties and impedance data were measured using impedance analyzer (HP4294A, Agilent, Palo Alto, CA, US). Impedance data were analyzed by a Z-view software. The temperature-dependent electrical performance tests were completed with the help of a temperature-controlled probe station (Linkam-HFS600E-PB2, London, UK).

## 3. Results

Figure 1 displays the X-ray diffraction (XRD) patterns of the CBTi-BFO films annealing at four different temperatures and standard JCPDS cards of the target phases. From Figure 1a, as the temperature is at a lower value of 450 °C, most of the diffraction peak of Aurivillius phases have not appeared yet or just a bump, indicating that some amorphous phases formed due to insufficient heat energy. Note that the CBTi-BFO films annealed at 500 and 550 °C show the (119) and (200) diffraction peaks accompanied with another peak with low intensity, which are consistent with the reported results of other CBTi films with Aurivillius phase, demonstrating a polycrystalline bismuth layered structure without any second phase [31,32,33]. As the annealing temperature continued to rise to 600 °C, apart from the Aurivilius phase, two diffraction peaks that do not belong to the characteristic of the bismuth layered ferroelectric appeared near 15° and 30°, which is due to the formation of the pyrochlore phase of Bi_2_Ti_2_O_7_ [34,35,36,37]. Generally, the pyrochlore phase can be formed frequently due to the bismuth element volatilize as the annealing temperature increases, resulting in the ratio of bismuth ions to titanium ions approaching 1:1. As shown in Figure 1b, the diffraction peak of (119) slightly shifts to a large angle with the increase of the annealing temperature, which may be caused by the release of surface residual stress [38,39].

The SEM images clearly display the surface and the cross-sectional morphologies of the CBTi-BFO thin films annealing at different temperatures. In Figure 2, the inset images reveal that the thickness of all samples is approximately 550 nm. Meanwhile, all CBTi-BFO thin films present compact and pore-free surface, which is favorable to energy storage performance. As shown in Figure 2a, when the film is annealing at 450 °C, relatively uniform fine grains can be noticed on the surface. As the temperature rises to 500 °C, the grains absorb heat energy and thus, increase uniformly (Figure 2b). In sharp contrast, the CBTi-BFO film annealing at 550 °C possesses different grains with a wide range of grain size from ~17 to ~70 nm in Figure 2c. As the annealing temperature rises up to 600 °C, the grain size further increases in the range of ~20 to ~85 nm (Figure 2d). The phenomenon can be ascribed to: (i) different grain shapes corresponding to different orientations [40]; (ii) an inhomogeneous nucleation and grain-growth at a high annealing temperature [41].

It is well known that thin film with a high dielectric constant (*ε*_r_) typically achieves tremendous recoverable energy density and energy efficiency [27,42]. Figure 3a presents the room-temperature frequency dependence of dielectric constant (*ε*_r_) and the dissipation factor (tan*δ*) of the CBTi-BFO thin films annealing at different temperatures. The *ε*_r_ value of each film slightly decreases with the frequency raising, and increases obviously with the annealing temperature increasing. The values of *ε*_r_ for the films annealing at 450 and 500 °C have slightly changed with the frequency increases, indicating that the samples annealed at these two temperatures have better frequency stability. Generally, the dielectric properties of ferroelectric film contain the intrinsic and extrinsic contributions, which could be influenced by different factors, such as the grain size, preferred orientation, and so on [43,44]. The reason that in the changes of *ε*_r_ with increasing annealing temperature may be due to the fact that the increased annealing temperature resulted in the increased grain size and reduced grain boundaries, leading to the enhancement of *ε*_r_ [27]. Moreover, the dielectric loss (tan*δ*) gradually increases with increasing frequency in all samples. Besides, all samples possess smaller loss (tan*δ* < 0.08) at 10 kHz.

Generally, *E*_b_ is analyzed by two parameter Weibull statistics, which is closely affected the energy storage performance of dielectric materials [45], as displayed in Figure 3b. Meanwhile, the *E*_b_ endurance of four films with different annealing temperatures are demonstrated in the inset of Figure 3b. The Weibull distribution of *E*_b_ can be expressed by the following formulas:(4)$$X_{i}=Ln\left(E_{i}\right)$$
(5)$$Y_{i}=Ln\left(-Ln\left(1-\frac{i}{n+1}\right)\right)$$
where *E*_i_ is the breakdown electric field for each sample, *i* signifies the number of test samples and *n* denotes the total number of test samples. Based on the Weibull distribution function, the mean *E*_b_ for each film can be obtained from the intersection of the fitted lines with the horizontal axis at Y*_i_* = 0. *β* represents Weibull shape parameter. It can be observed that *E*_b_ increases rapidly with the annealing temperature decreasing, as shown in the inset of Figure 3b. The average breakdown strength of the dielectric film increases from 3241 kV·cm^−1^ to 3984 kV·cm^−1^, with the annealing temperature decrease from 600 to 450 °C. The enhancement of dielectric breakdown strength can be ascribed to the following factor. It is well-known that electric breakdown field strength is inversely proportional to the grain size (*G*), which can be manifested by the following formula:(6)$$E_{b}\propto {\left(G\right)}^{-a}$$
where *a* is the exponent values, being in the range of 0.2–0.4 [6,46]. It can be seen that the dielectric breakdown strength increases with the decreased grain size, which is aligned with the results in Figure 2 and Figure 3b. That is owing to grain boundaries producing depletion regions similar to Schottky barriers located in semiconductor interfaces. Then, the grain boundaries depletion layers establish important barriers for the cross-transport of ionic and electronic charges [47]. Therefore, the *E*_b_ for CBTi-BFO film annealing at 450 °C is superior to the films that have higher annealing temperature. The slope parameter *β*, related to the scatter of *E*_b_ data, increases from 12.88 to 17.71 with the annealing temperature decrease from 600 °C to 450 °C, indicating an enhancement in dielectric reliability by annealing temperature decreasing. At the same time, the *β* of all films based on the linear fitting is higher than 12, suggestive of all samples possessing high reliability.

Figure 3c represents the leakage current of all films, which are measured by applying 0–185 kV·cm^−1^ to the electrodes. The CBTi-BFO films annealed at 450 and 500 °C illustrate well insulating properties with leakage current densities < 7 × 10^−7^ A cm^−2^ under an applied electric field of 185 kV·cm^−1^. The CBTi-BFO films annealed at 550 and 600 °C exhibit higher leakage currents, which is ascribed to the greater grain size and the simultaneously declined number of the grain boundary.

Presented in Figure 3d are bipolar polarization-electric filed (*P-E*) loops of the CBTi-BFO films annealing at different temperatures applying the electric field of 2500 kV·cm^−1^ at frequency of 10 kHz. With the annealing temperature decreasing, a monotonous decrease of polarization (*P*_max_) is observed, from 33 μC cm^−2^ of 450 °C to 63 μC cm^−2^ of 600 °C.

Figure 3e shows *P-E* loops of CBTi-BFO films near their respective *E*_b_ annealing at different temperatures. It can be seen that the CBTi-BFO thin film annealing at 600 °C presents a practically saturated *P-E* loop with a remnant polarization *P*_r_ of 20 μC/cm^2^, which is lower than previous reports [32,33], suggesting that the derived CBTi-BFO thin films have somewhat discrepant ferroelectric properties. For the CBTi-BFO thin films annealing at lower temperatures, the obtained *P-E* hysteresis loops exhibit slim shape, which can be ascribed to the function of lower leakage current and small crystallite size [48,49,50]. Figure 3f presents the corresponding energy storage parameters of *W*_rec_, *W*_loss_ and *ƞ* determined from *P-E* loops about all CBTi-BFO films. The value of *W*_rec_ and *ƞ* for the CBTi film annealing at 500 °C, respectively, reaches 82.8 J·cm^−3^ and 78.3% due to an integration of remarkable *E*_b_ of 3596 kV·cm^−1^ and a large polarization disparity of 52.3 μC cm^−2^ (inset of Figure 3e). It is well known that the dielectric film can induce outstanding energy performance due to high *P*_max_, low *P*_r_ and high electric breakdown strength. The lower annealing temperature leads to more slender electric hysteresis loops and lower *W*_loss_. Meanwhile, *E*_b_ is enhanced with the decrease of the annealing temperature.

Figure 4a–d show the Nyquist plots of films measured in the frequency range of 100 Hz to 1 MHz at a series of temperatures (225–300 °C). As is known to all, a low-frequency arc means the dielectric response of grain boundaries, while a high-frequency arc means the dielectric response of grains. One semicircular arc is observed for the samples. As the grain boundary dielectric relaxation of the material mainly contributes to the total impedance, only the semi-circular arc corresponding to the grain boundary response is observed. The intercept of impedance semicircular arcs on the Z’-axis can represent the total resistance (*R*_b_) values of the film. Clearly, the value of *R*_b_ gradually decreased with the increasing measured temperature in each film, showing a negative temperature characteristic. This phenomenon can be attributed to the fact that the mobility of the space charge becomes easier, and more charge carriers will accumulate at the grain boundaries with the increasing measured temperature, thus resulting in increased electrical conductivity and decreased grain boundary resistance [51,52].

In order to compare the impedance differences of the four samples in more detail, we separately take out the impedance diagrams tested at 300 °C and showed them in Figure 5a. It can be seen that *R*_b_ of CBTi-BFO films gradually increases with a decreasing annealing temperature, indicating that the charge carrier concentration decreases in low-temperature annealed samples [53].

Generally, the energy required for carriers to cross the energy barrier is called conductive activation energy (*E*_a_), which can be calculated by Arrhenius formula:(7)$$\sigma =\sigma_{0}\exp\left(-E_{a}/k_{B}T\right)$$
where *T*, *σ*_0_, *E*_a_ and *k*_B_ are the measuring temperature on the Kelvin scale, preexponential factor, activation energy and Boltzmann constant, respectively. The plots of ln(*σ*) as a function of 1000/T for the films and linear fittings are shown in Figure 5b. The activation energies (*E*_a_) obtained from the slopes of fitting lines for films annealing at 450, 500, 550, 600 °C are estimated to be 1.21, 0.84, 0.68, 0.66 eV, respectively. Generally, the higher value of activation energy means fewer defects exist in the film [54]. Thus, we can deduce that the sample with the lowest annealing temperature has the least defects. Combining the above two points, it can be seen that CBTi-BFO with lowest annealing temperature possesses the smallest charge carrier concentration and fewer defects, which contribute greatly to the observed highest intrinsic breakdown field in this film.

For practical application, the requirement of reliability and stability is emphasized, such as under high/low temperature, high/low frequency and long-term working environments. Thus, in view of the CBTi-BFO film annealing at 500 °C possessing large *W*_rec_ and *ƞ* among all samples, further investigations are carried out on it under an electric field of 2500 kV·cm^−1^. Firstly, the temperature stability of the CBTi-BFO film annealing at 500 °C is measured. The bipolar *P-E* loops of the film measured from relatively low temperature of −25 °C to ultrahigh temperature of 200 °C are displayed in Figure 6a. For the purpose of expressing the variation of polarization more clearly, the unipolar hysteresis loop diagrams’ dependence on temperature is drawn in Figure 6d. It can be seen that the *P*_max_ value varies slightly from 42.62 to 45.89 μC cm^−2^. Correspondingly, *W*_rec_ values are slightly increased by 2% from 44.88 to 45.84 J·cm^−3^ and *ƞ* is reduced by 6% from 80.04 to 75.19% with temperature increasing, as shown in Figure 6g. The aforesaid energy storage performance of the film shows good temperature stability, which is adequate to meet the demands of applications in extreme environments (capacitors used in underground industrial instruments need to work at temperatures higher than 150 °C and the inverter must work at about 140 °C in HEV) [55,56,57]. Besides, frequency dependence of the energy storage behavior is also researched at 2500 kV·cm^−1^. Figure 6b, e are the bipolar and unipolar hysteresis loops measured at the frequency range of 500 Hz to 20 kHz, respectively; and the corresponding energy performance *W*_rec_ and *ƞ* are shown in Figure 6h. Clearly, the *P-E* loop can maintain a slim feature and no discernible decline in energy storage performance can be discovered. Even though the frequency rises from 500 Hz to 20 kHz, the *W*_rec_ and *ƞ* values slightly drop from 48.71 to 43.80 J·cm^−3^ and 83.94 to 77.47%, respectively, indicating that a good frequency stability can be realized. Furthermore, to assess the long-term charging–discharging stability of dielectric capacitors, the fatigue endurance should be investigated. The *P-E* loops of CBTi-BFO film annealing at 500 °C over 10^7^ charge–discharge cycles are exhibited in Figure 6c, f. It can be seen that there is no obvious change in the hysteresis loops. The corresponding *W*_rec_ and an excellent *ƞ* present a negligible degradation with 1% and 0.3% as shown in Figure 6i. The observed superior antifatigue feature is closely related to the (Bi_2_O_2_)^2+^ layer in the Aurivillius phase, which has a good insulating effect inhibiting the flow and generation of leakage charges during the repeated polarization process. Thus, the Aurivillius phase can more effectively suppress the electrical breakdown during the fatigue process and improve the anti-fatigue performance.

Figure 7 summarizes a serious of results on the energy storage performance of Aurivillius ferroelectric films [12,13,27,28,58,59,60]. In this study, the CBTi-BFO thin film annealing at 500 °C possesses a relatively high *ƞ* (~78%) superior than BaLa_0.2_Bi_3.8_Ti_4_O_15_ (BLBT~60%) and Bi_3.25_La_0.75_Ti_3_O_12_/BiFeO_3_/Bi_3.25_La_0.75_Ti_3_O_12_ (BLT/BFO/BLT~74%), but inferior to Ba_2_Bi_4_Ti_5_O_18_ (BBT~92%), 0.6BaTiO_3_-0.4Bi_3.25_La_0.75_Ti_4_O_12_ (0.6BT-0.4BLT~84%), CaBi_2_Nb_2_O_9_ (CBNO~82%), Sr_2_Bi_4_Ti_5_O_18_ (SBT~81%), Bi_3.25_La_0.75_Ti_3_O_12_-BiFeO_3_ (BLT-BFO~80%). In contrast, its *W*_rec_ clearly outperforms the surveyed dielectric systems with a high *ƞ*. That is, the CBTi-BFO thin film annealing at 500 °C has excellent comprehensive properties, namely, a good balance between *W*_rec_ and *η*. Meanwhile, the film has a great *E*_b_ (~3596 kV·cm^−1^) at a high level. Taken together, the CBTi-BFO thin film annealing at 500 °C is a good candidate for application in energy storage devices.

## 4. Conclusions

In this work, a series of CBTi-BFO thin films are prepared by chemical solution deposition under different annealing temperatures. We found that by optimizing the microstructures and crystalline structure of films via adjusting the annealing temperature, the electrical properties and energy storage performance can be greatly improved, especially in the film with a relatively lower annealing temperature. In the annealing temperature of 500–550 °C, CBTi-BFO thin films demonstrates a polycrystalline bismuth layered structure without any second phase. The CBTi-BFO with the lowest annealing temperature has the smallest carrier concentration and fewer defects, which greatly contribute to the highest intrinsic breakdown field of this film. Here, ultrahigh energy storage density of *W*_rec_ (~82.8 J·cm^−3^) and *ƞ* (~78.3%) are achieved in the CBTi-BFO film that annealed at 500 °C. The excellent energy storage performance can be ascribed to its uniform fine grain size. The film also shows superior thermal stability (from −25 to 200 °C), frequency stability (from 500 Hz to 20 kHz) and fatigue endurance (after 10^7^ switching cycles). In a word, annealing temperature plays an important part in performance tuning, which is a vital factor needed to be considered for preparing thin film capacitors with high energy storage characteristics.

## Figures and Tables

**Figure 1 nanomaterials-12-00730-f001:**
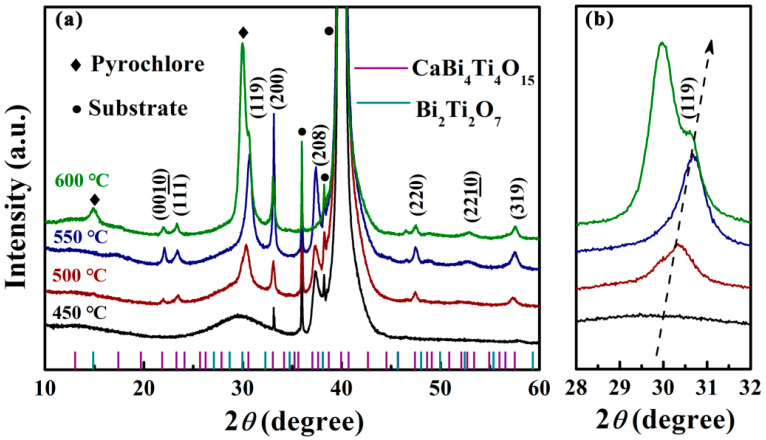
(**a**) X-ray diffraction (XRD) patterns in the 2*θ* range of 10–60° of the CBTi-BFO films annealed at various temperatures. (**b**) Enlarged XRD patterns of the diffraction peaks of 2*θ* at around 30°.

**Figure 2 nanomaterials-12-00730-f002:**
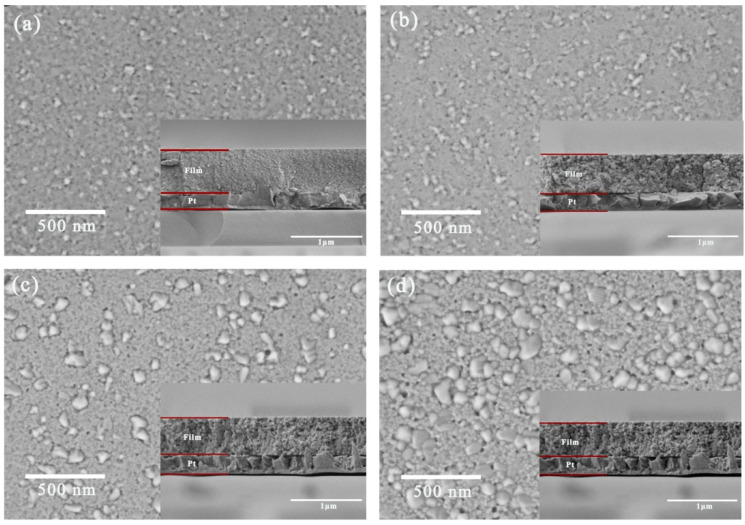
The scanning electron microscopic (SEM) images of the CBTi-BFO thin films annealed at (**a**) 450 °C, (**b**) 500 °C, (**c**) 550 °C and (**d**) 600 °C, and the inset shows their corresponding cross-sectional micrographs.

**Figure 3 nanomaterials-12-00730-f003:**
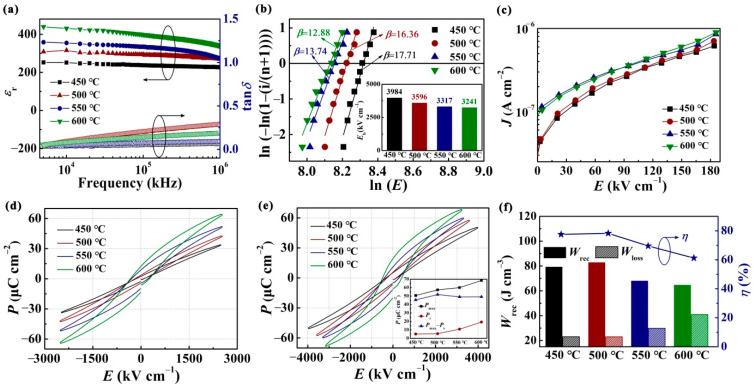
CBTi-BFO films annealed at 450, 500, 550 and 600 °C: (**a**) Frequency dependence of dielectric properties. (**b**) Weibull distributions dielectric breakdown strengths. (**c**) Leakage current density as a function of applied electric filed. (**d**) The bipolar *P-E* loops at 2500 kV·cm^−1^. (**e**) The bipolar *P-E* loops around *E*_b_, The inset shows the variation of *P*_max_, *P*_r_, and *P*_max_-*P*_r_ as a function of annealing temperature. (**f**) The variations of *W*_rec_, *W*_loss_ and *ƞ* values.

**Figure 4 nanomaterials-12-00730-f004:**
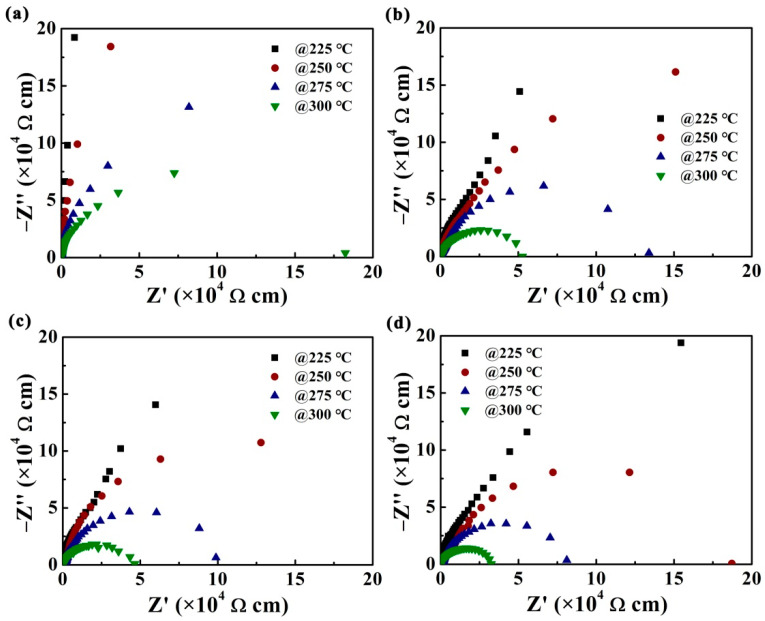
Nyquist diagrams of complex-plane of the CBTi-BFO films with annealing temperature of (**a**) 450, (**b**) 500, (**c**) 550, (**d**) 600 °C measured at 20 V under four measured temperatures.

**Figure 5 nanomaterials-12-00730-f005:**
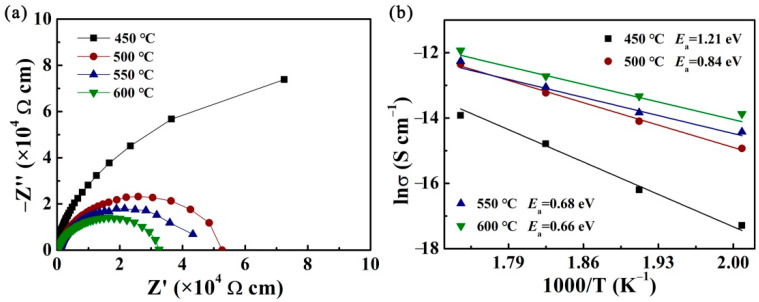
(**a**) The impedance and fit result of CBTi-BFO films measured at 300 °C. (**b**) Arrhenius plots of CBTi-BFO films annealing at different temperatures.

**Figure 6 nanomaterials-12-00730-f006:**
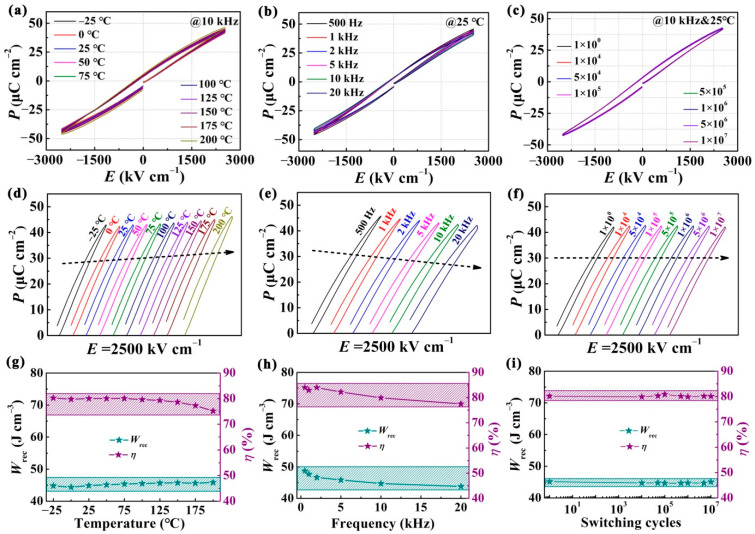
CBTi-BFO thin film annealed at 500 °C: (**a**–**c**) bipolar *P-E* hysteresis loops measured at different temperatures/frequencies/switching cycles; (**d**–**f**) unipolar *P-E* hysteresis loops of CBTi-BFO films at different temperatures/frequencies/switching cycles; (**g**–**i**) the changes of *W*_rec_ and *ƞ* depending on temperature/frequencies/switching cycles.

**Figure 7 nanomaterials-12-00730-f007:**
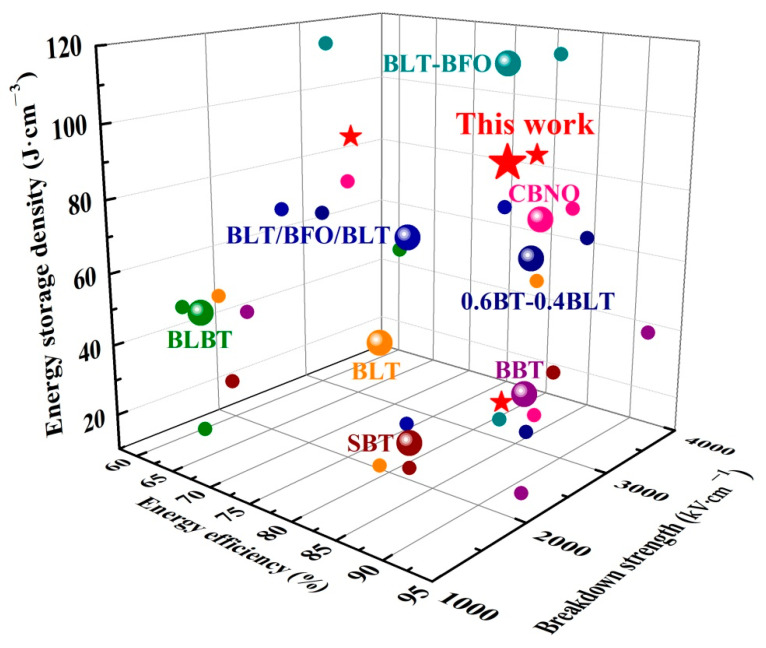
Summary of recently reported the core parameters of *E*_b_, *W*_rec_ and *ƞ* for energy storage properties of representative Aurivillius ferroelectrics film [12,13,27,28,58,59,60].

## Data Availability

Not applicable.

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
