# Peer review of "Excellent Energy Storage Performance in Bi(Fe0.93Mn0.05Ti0.02)O3 Modified CaBi4Ti4O15 Thin Film by Adjusting Annealing Temperature"

_nanomaterials, 2022, doi:10.3390/nano12050730_

Round 1
Reviewer 1 Report
The study sounds interesting. It can be published after some minor points have been checked and reflected.
1) Why did you choose CBTi among many compounds? Please present the reason in the manuscript.
2) Line 111- What is the designed thickness? Why was the thickness determined?
3) Figure 1 – Please indicate the JCPDS of the target phase on the XRD graph together.
4) In figure3(e), the Pm-Pr value of the sample annealed at 450°C is smaller than those annealed at higher temperatures. However, the sample annealed at 450°C shows a larger Wrec value in figure3(f). Please suggest the reason. (In line 54, it is said that Wrec can be improved by increasing the difference between Pm and Pr, 54 and the electric breakdown strength.)
5) Line 233 – Please add the temperature conditions of the test results to the manuscript.
6) In the conclusion, authors are encouraged to suggest additional data related to the research results.
Reviewer 2 Report
In the present report, the authors constructed a new lead-free bismuth layer-structured ferroelectric thin films of CaBi4Ti4O15- Bi(Fe0.93Mn0.05Ti0.02)O3 (CBTi-BFO) were prepared via chemical solution deposition. The effect of annealing temperature on the energy storage performances of a series of thin films was investigated. Especially, the CBTi-BFO film annealed at 500°C shows a high recoverable energy density of 83 J/cm3 and efficiency of 78%, which can be attributed to the very slim hysteresis loop and a relatively high electric breakdown strength. This manuscript provided some valuable information and the content is very significant in this field. However, I recommended a major revision of the article from its present form before it can be published in nanomaterials. Some specific comments are as follows:
- The abstract and conclusion sections should be a specific and scientific approach.
- In the introduction section, the authors should expound on the research significance of the present work.
- The authors should explain the novelty of the present report?
- The authors should provide a schematic representation of the formation mechanism.
- The abbreviations and labels are not properly organized throughout the manuscript.
- What is the pH of the reaction solution? The pH of the solution normally varies from precursor to precursor. The authors must justify the selection of pH, temperature, and time.
- The authors should include error bars in all results for more accuracy.
- All figures are of very poor quality. The authors should provide high-quality images.
- What is the key factor affecting energy storage performance?
- In the current state, there are more typographical errors and the language should be improved. Therefore, the authors are advised to recheck the whole manuscript for improving the language and structure carefully.
Round 2
Reviewer 2 Report
The manuscript can be acceptable form.